# Oscillation Inversion: Understand the structure of Large Flow Model through the Lens of Inversion Method

## Abstract

We investigate oscillation phenomena observed in inversion methods applied to large text-to-image diffusion models, particularly the "Flux" model. Using a fixed-point-inspired iteration method to invert real-world images, we find that the solution does not converge but instead oscillates between distinct clusters. Our results, validated both on real diffusion models and toy experiments, show that these oscillated clusters exhibit significant semantic coherence. We propose that this phenomenon arises from oscillatory solutions in dynamic systems, linking it to the structure of rectified flow models. The oscillated clusters serve as local latent distributions that allow for effective semantic-based image optimization.We provide theoretical insights, linking these oscillations to fixed-point dynamics and proving conditions for stable cluster formation and differentiation in flow models.

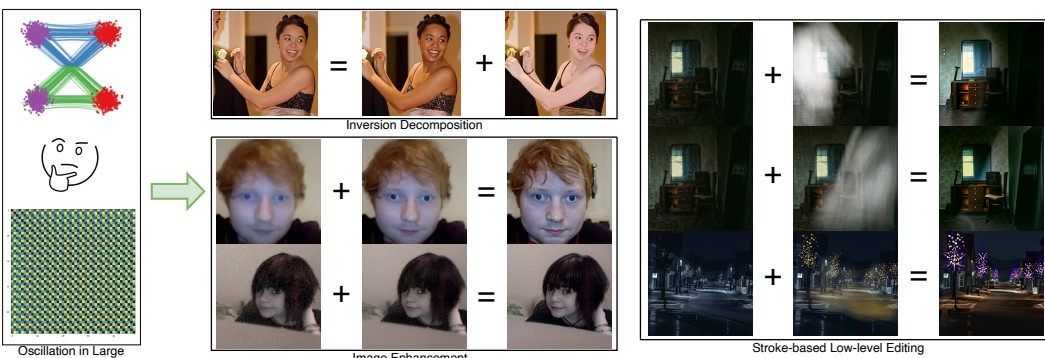

Figure 1: Oscillation Inversion is a phenomenon observed in large flow models that can serve as a low-level image editing technique. It enables image enhancement and visual prompt-based low-level editing.

## 1 Introduction

Very recently, large text-to-image diffusion models utilizing rectified flow ( Wang et al. (2024)), like the Flux model from Black Forest Labs, have demonstrated exceptional performance in generating high-quality images with rapid sampling. However, the underlying latent structure of rectified flow-based models presents unique challenges, as it differs fundamentally from the layered manifold structure of DDPMs ( Ho et al. (2020)) shaped by a parameterized Markov chain. This distinction makes previous inversion techniques, such as DDIM inversion ( Song et al. (2020a)), and editing methods like SDEdit ( Meng et al. (2021)), less viable. Therefore, adopting a new perspective for understanding and navigating the latent space of these flow-based models is essential for enabling more effective inversion and image manipulation strategies.

When attempting to invert real-world images using fix-point iteration methods in rectified flow-based models, we observe that the sequence of iterates does not converge to a single point but

instead oscillates between several clusters. These clusters are semantically meaningful and can be leveraged for image optimization and editing tasks. This behavior contrasts with the fixed-point methods used in DDIM, which are primarily designed to mitigate numerical accumulation errors at each step, ensuring smooth convergence to a single, stable solution. The oscillatory nature observed in rectified flow-based models, however, opens up new possibilities for iterative refinement and enhanced flexibility in image inversion tasks.

To investigate this, we first propose **Oscillation Inversion**, a method that uses fixed-point iteration to directly establish a one-to-one mapping between noisy latents at a given timestep and the corresponding encoded image latent. The inverted latents oscillate among several clusters, which can serve as local latent distributions, facilitating effective semantic-based image optimization. Additionally, we generalize this fixed-point method in three ways for broader downstream applications: **1) Group Inversion**: We invert a group of images simultaneously, rather than a single image, enabling semantic guidance and blending across images. **2) Finetuned Inversion**: By controlling the oscillation direction, we provide a mechanism for customized, user-driven manipulation of the images. **3) Post-Inversion Optimization**: After inversion, we perform optimization, and analyze the differentiable structure induced by the separated sub-distributions created by oscillation inversion. These extensions make Oscillation Inversion a versatile tool for various image manipulation tasks.

The main contributions of this work are as follows:

- We propose Oscillation Inversion to facilitate one-step inversion to any timestep in rectified flow-based diffusion models for the semantic manipulation of latents. Additionally, we present three extensions that enable diverse user inputs for real-world applications.

- Extensive experiments on various downstream tasks, such as image restoration and enhancement, stroke based make up transfer, validate our theoretical findings and demonstrate the effectiveness of our method on both perceptual quality and data fidelity.

## 2 RELATED WORKS

**Flow Model.** Diffusion models Rombach et al. (2022) Saharia et al. (2022) Ramesh et al. (2022) generate data by a stochastic differential equation (SDE)-based denoising process and probability flow ordinary differential equations (ODE) Song et al. (2020b) Lipman et al. (2022) Lipman et al. (2022) Salimans & Ho (2022) Song et al. (2023) improves sampling efficiency by formulating the denoising process into a ODE-based process. However, probability flow ODE-based methods suffer from the computational expense of denoising via numerical integration with small step sizes. To address these issues, some simulation-free flow models have emerged, e.g. flow matching Lipman et al. (2022) and rectified flow Liu et al. (2022). Flow matching introduces a training objective for continuous normalizing flows Chen et al. (2018) to regress the vector field of a probability path. Rectified flow learns a transport map between two distributions through constraining the ODE to follow the straight transport paths. Since the latent structure of flow models differs fundamentally from the layered manifold structure of Denoising Diffusion Probabilistic Models (DDPMs) Ho et al. (2020), it is valuable to explore the intrinsic characteristics of the flow models' latent space.

**Diffusion-based Inversion.** The rise of diffusion models Rombach et al. (2022) Saharia et al. (2022) Ramesh et al. (2022) has unlocked significant potential of inversion methods for real image editing, which are primarily categorized into Denoising Diffusion Probabilistic Models (DDPM Ho et al. (2020))-based Wu & De la Torre (2023) Huberman-Spiegelglas et al. (2024) and Denoising Diffusion Implicit Models (DDIM Song et al. (2020a))-based methods Pan et al. (2023b) Garibi et al. (2024) Li et al. (2024) Meiri et al. (2023a). While DDPM-based methods yield impressive editing results, they are hindered by their inherently time-consuming and stochastic nature, due to the random noise introduced across a large number of inversion steps Wu & De la Torre (2023) Huberman-Spiegelglas et al. (2024). DDIM-based methods utilize the DDIM sampling strategy to enable a more deterministic inversion process, substantially reducing computational overhead and time. However, the linear approximation behind DDIM often leads to error propagation, resulting in reconstruction inaccuracy and content loss, especially when classifier-free guidance (CFG) is applied Mokady et al. (2023). Recent approaches, Wallace et al. (2023) Mokady et al. (2023) Pan et al. (2023b) Miyake et al. (2023) Han et al. (2023) Hong et al. (2024), address these issues by aligning the diffusion and reverse diffusion trajectories through the optimization of null-text tokens Mokady et al. (2023)

or prompt embeddings Han et al. (2023) Miyake et al. (2023). EDICT Wallace et al. (2023) and BDIA Zhang et al. (2023) introduce invertible neural network layers to enhance computational efficiency and inversion accuracy, though these methods suffer from notably longer inversion times. To tackle this, recent works Meiri et al. (2023a) Pan et al. (2023b) Garibi et al. (2024) Li et al. (2024) have adopted fixed-point iteration for each inversion step, mitigating numerical error accumulation and ensuring smooth convergence to a single, stable solution. Interestingly, when applied to rectified flow-based methods, the sequence of fixed-point iterates oscillates between several semantically meaningful clusters, presenting significant potential for downstream applications.

## 3 OSCILLATION INVERSION

### 3.1 PRELIMINARY

#### 3.1.1 RECTIFIED FLOW

Rectified flow (Liu et al. (2022)) is a novel generative approach that facilitates smooth transitions between two distributions, denoted $\pi_0$ and $\pi_1$, by solving ordinary differential equations (ODEs). Specifically, for $X_0 \sim \pi_0$ and $X_1 \sim \pi_1$, the transition between $x_0$ and $x_1$ is defined through an interpolation given by $X_t = (1 - t)X_0 + tX_1$ for $t \in [0, 1]$. Liu et al. (2022) demonstrated that, starting from $Z_0 \sim \pi_0$, the following ODE can be used to obtain a trajectory that preserves the marginal distribution of $Z_t$ at any given time $t$:

$$\frac{dZ_t}{dt} = v^X(Z_t, t), \quad \text{where } v^X(x, t) := \mathbb{E}[X_1 - X_0 \mid X_t = x]. \tag{1}$$

The solution of $v^X$ in Eq. (1) is obtained by optimizing the following loss via stochastic coupling sampling $(X_0, X_1) \sim (\pi_0, \pi_1)$ and $t \sim \text{Uniform}([0, 1])$,

$$v^X = \arg\min_v \mathbb{E}\left[\left\|(X_1 - X_0) - v(tX_1 + (1 - t)X_0, t)\right\|^2\right]. \tag{2}$$

### 3.2 METHOD

In this section, we first formulate the inversion problem for rectified flow-based models (Sec.3.2.1). To address this, we introduce Oscillation Inversion, which constructs one-step inversion using fixed-point iteration (Sec.3.2.2). We then propose fine-tuned inversion to enable controllable latent structures (Sec.3.2.3), followed by post-inversion optimization for further image refinement (Sec.3.2.4). The concept behind our general method is depicted in Figure 2.

#### 3.2.1 INVERSION PROBLEM

In practice, the large flow model in the context of generative modeling operates within the latent space of a Variational Autoencoder (VAE) (Kingma (2013)), utilizing an encoder $E : \mathbb{R}^d \to \mathbb{R}^n$ and a decoder $D : \mathbb{R}^n \to \mathbb{R}^d$. The sampling process begins from Gaussian noise $z_T \sim \mathcal{N}(0, \mathbf{I})$, and the latent variable is progressively refined through a sequence of transformations. The forward generative process is defined by the following iterative formula starting from $t = T$ all the way back to $t = 0$:

$$z_{t-1} = z_t + (\sigma_{t-1} - \sigma_t) v_\theta(z_t, \sigma_t), \tag{3}$$

where $v_\theta(z_t, \sigma_t)$ represents the learned velocity field parameterized by a transformer with weights $\theta$, and $\sigma_t$ is a monotonically increasing time step scaling function depending on time $t$ with $\sigma_0 = 0$ and $\sigma_T = 1$. Here, $T$ denotes the total number of discredited timesteps. The final latent variable, $z_0$, is the output ready for decoding.

The inversion problem involves seeking for the initial noise $z_T$ given an observed pixel image $I$ with corresponding latent encoding $y \in \mathbb{R}^d$, such that generating from $z_T$ using the flow model described above allows us to either reconstruct $y$ or apply desired modifications to it.

However, the gradual process of sampling from Gaussian noise to the original $y$ diminishes the advantage of GAN-like one-step mappings for direct latent space optimization. To address this,

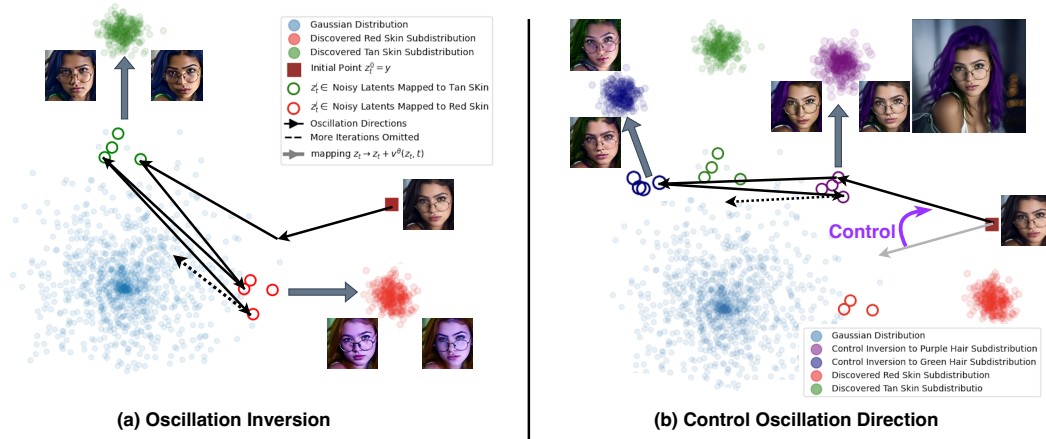

(a) Oscillation Inversion          (b) Control Oscillation Direction

Figure 2: In the left figure (a), fixed-point iteration causes oscillation, leading to subdomains with opposite features in the case of the brown-skinned girl, resulting in more tan and red tones. In the right figure (b), we demonstrate how this oscillation can be customized to achieve desired separations, such as changes in hair color.

unlike tackling with the initial noise at $t = T$, we introduce the assumption that, at a selected intermediate timestep $t_0$, there exists a direct one-step mapping from the noisy latent at timestep $t_0$ to the clean latent at timestep 0 via a "jumping" transformation.

More specifically, assuming $y$ is the latent code to recover, we aim to figure out the *intermediate* latent code $z_{t_0}$ satisfying

$$z_{t_0} + (\sigma_0 - \sigma_{t_0}) v_\theta(z_{t_0}, \sigma_{t_0}) = y. \tag{4}$$

Solving Eq. (4) is non-trivial, and in the following sections, we will describe how we find a set of approximated solutions using iterative method and analyze its oscillating properties.

### 3.2.2 OSCILLATIONS INVERSION

To address the inversion problem, we employ a fixed-point iteration method to approach the solution of Eq. (4). Instead of directly seeking a point $z_{t_0}$ such that applying the one-step generative process as described in the left side of (4) yields the target latent $y$, we define an iterative process that refines our approximation of the inverted latent code. We define the fixed-point iteration as:

$$z_{t_0}^{(k+1)} = y - (\sigma_0 - \sigma_{t_0}) v_\theta(z_{t_0}^{(k)}, \sigma_{t_0}), \tag{5}$$

with the initial condition $z_{t_0}^{(0)} = y$. The sequence $\{z_{t_0}^{(k)}\}_{k=0}^{\infty}$ represents successive approximations of the inverted latent code at timestep $t$.

Rather than converging to a single point as suggested by Banach's Fixed-Point Theorem (Banach (1922)), we empirically observed that the sequence $\{z_{t_0}^{(k)}\}_{k=0}^{\infty}$ generally oscillates among several clusters in the latent space. Each cluster corresponds to a semantically concentrated region that shares similar low-level features. This oscillatory behavior can be harnessed to explore different variations of the input image, providing a richer inversion that captures multiple aspects of the data.

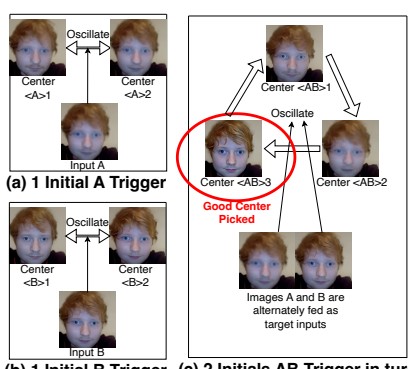

(a) 1 Initial A Trigger  (b) 1 Initial B Trigger  (c) 2 Initials AB Trigger in turn

Figure 3: This is an example of group inversion, where the high-quality distribution is triggered by two degenerate distributions. We also apply this method to large-scale experiments in Section 4.1.

Building upon the fixed-point method, we introduce **group inversion** that can trigger more stable oscillation phenomena by inverting a set of images simultaneously in a periodic fashion as shown in

Figure 3. Suppose we obtain their corresponding latent encodings $y_1, \ldots, y_m$ from a collection of images $I_1, \ldots, I_m$ using the VAE encoding. Given a sequence $b_1, \ldots, b_N \in \{1, \ldots, m\}$, we could perform the iteration on the group:

$$z_{t_0}^{(k+1)} = y_{b_{(k \bmod N)}} - (\sigma_0 - \sigma_{t_0}) v_\theta(z_{t_0}^{(k)}, \sigma_{t_0}), \tag{6}$$

with initial conditions $z_{t_0}^{(0)} = y_{b_1}$. By inverting the images together, we enable interactions between their latent representations during the iteration process. Our experimental findings indicate that this collective inversion induces more diverse oscillatory clusters in the latent space. Each cluster center represents a semantic blending of the input group of images, capturing shared attributes and features across the group. This process could provide a basis for semantic guidance and blending, potentially supporting tasks such as style transfer, identity transfer, and object modification (e.g., adding or removing objects) by exploring the latent space influenced by the oscillations. Also, the oscillation phenomenon observed in the flow model trained on a toy distribution transitions from a large central Gaussian to a mixture of four smaller Gaussians, as showed in Figure 6 This behavior aligns closely with the results from our experiments on larger models.

### 3.2.3 FINTUNED INVERSION

While the fixed-point iteration method introduced earlier reveals oscillations that segment the latent space into separate clusters, these clusters are not directly controllable.

To overcome this limitation, we propose **finetuned inversion**, a simple fine-tuning step that adjusts the inversion direction, allowing the separated clusters to align with customized semantics. This approach can also induce more diverse oscillatory separations in the latent space.

Given an input image $I$ with its encoded latent representation $y$, we consider the image $\tilde{I}$ modified from $I$ that reflects desired editing, such as alterations made using off-the-shelf masking and in-painting models or customized doodling edits. For example, if original $I$ is an image of a girl with black hair, $\tilde{I}$ could be a roughly edited version where the girl's hair is doodled to appear purple. Importantly, the edited image $\tilde{I}$ does not have to be photo-realistic nor perfect.

Encoding the edited image $\tilde{I}$ to obtain $\tilde{y}$, our goal is to fine-tune the parameters $\theta$ of the velocity field network $v_\theta$ so that the inversion process aligns with the desired modifications. The fine-tuning optimization problem could be formulated as below.

$$\underset{\theta}{\text{minimize}} \, \mathcal{L}_{\text{finetune}} = \left\| v_\theta(y, \sigma_t) - v^{\text{pt}}(\tilde{y}, \sigma_t) \right\|_2^2, \tag{7}$$

where $v_\theta$ is the velocity field parameterized by $\theta$ that we aim to finetune, and $v^{\text{pt}}$ is the pretrained velocity field with frozen weights.

Through force aligning $v_\theta(y)$ and $v^{\text{gt}}(\tilde{y})$ via optimizing over $\theta$, we effectively put strong guidance on the inversion direction so that the resulting oscillatory clusters obtained as described in Sec. 3.2.2 would be pulled towards the semantics specified by the reference image $\tilde{I}$.

### 3.2.4 POST-INVERSION OPTIMIZATION

Following the fixed-point iterations and optional fine-tuning, we perform a **post-inversion optimization** to refine the found latent code $z_t$ further. Unlike previous approaches that aim at enhancing image quality, our optimization focuses on utilizing the sub-distribution formed by the oscillatory clusters to guide the latent code towards clusters with desired semantics while maintaining stylistic consistency with the cluster.

After identifying the sub-distribution characterized by the points in the clusters obtained from the oscillations, we

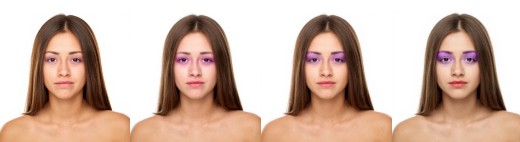

**Center of Sub Cluster 1 after Fintuning Step 1, 3, 5, 15**

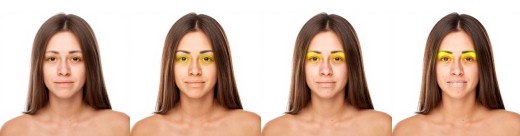

**Center of Sub Cluster 2 after Fintuning Step 1, 3, 5, 15**

Figure 4: Gradual evolution of two oscillating distributions occurs under the influence of visual prompts during fine-tuning. We utilize this method in the experiments discussed in Section 4.2.

use these points as reference samples in a Gaussian Process (GP) framework to serve as regularization in our optimization loss. This approach ensures that the optimized latent code retains the same style and features as the sub-distribution while allowing gradual evolution guided by a customized loss function.

The optimization problem is formulated as:

$$\underset{z_{t_0}}{\text{minimize}} \; \mathcal{L}_{\text{rgb}} \circ D\big(z_{t_0} + (\sigma_0 - \sigma_{t_0}) v_\theta(z_{t_0}, \sigma_{t_0})\big) + \beta \, \mathcal{L}_{\text{GP}}(z_{t_0}), \tag{8}$$

where $\circ$ represents function composition operator while $D(\cdot)$ is the VAE decoder that converts the latent code back to the pixel space, $\mathcal{L}_{\text{rgb}}$ is a customized loss function defined directly on the decode pixel image space designed based on the specific application or desired attributes. Another loss $\mathcal{L}_{\text{GP}}(\cdot)$ is the Gaussian Process regularization term that encourages $z_{t_0}$ to stay close in distribution to the sub-distribution formed by the oscillatory cluster. $\beta$ is a scalar hyperparameter that balances the influence of the customized loss and the GP regularization.

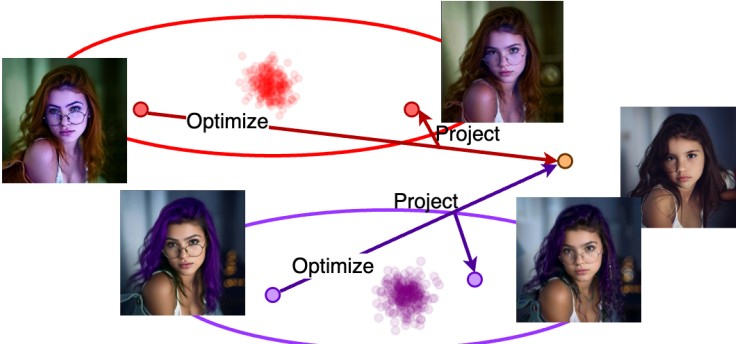

Figure 5: Illustration of how our Gaussian Regularization Domain Preserving method works. The red skin and purple hair domains are optimized toward minimizing photo loss, resulting in an image that retains these domains while appearing younger.

The Gaussian Process regularization term $\mathcal{L}_{\text{GP}}(z)$ is computed using a Radial Basis Function (RBF) kernel to measure similarity between $z_{t_0}$ and the reference points from the sub-distribution. Specifically, the GP loss is defined as:

$$\mathcal{L}_{\text{GP}}(z) := k(z, z) - \frac{2}{N} \sum_i k(z, z_i') + \frac{1}{N^2} \sum_{i,j} k(z_i', z_j'), \tag{9}$$

where $k(a, b) := \exp\big(-\frac{\|a-b\|^2}{2l^2}\big)$ is the RBF kernel with scale $l$, $\{z_i'\}_{i=1}^{N}$ are $N$ reference latent codes from the oscillatory cluster (sub-distribution).

This GP loss is derived from the Maximum Mean Discrepancy (MMD) measure, which quantifies the difference between the distribution of $z_t$ and the sub-distribution represented by $\{z_i'\}$. By minimizing $\mathcal{L}_{\text{GP}}(z_{t_0})$, we encourage $z_{t_0}$ to share similar statistical properties with the cluster, thus maintaining stylistic and feature consistency.

In the overall optimization, the first term $\mathcal{L}_{\text{rgb}}$ guides the latent code towards satisfying specific goals or attributes defined by the application, such as emphasizing certain visual features or styles in the decoded image. The GP regularization term ensures that these changes remain coherent with the characteristics of the sub-distribution, preventing the optimization from deviating too far from the latent space regions that correspond to realistic and semantically meaningful images.

By integrating these two components, we achieve a balance between customizing the output according to desired specifications and preserving the inherent style and features of the sub-distribution identified through oscillation inversion. This method allows for controlled manipulation of the generated images while maintaining high fidelity and coherence with the original data manifold.

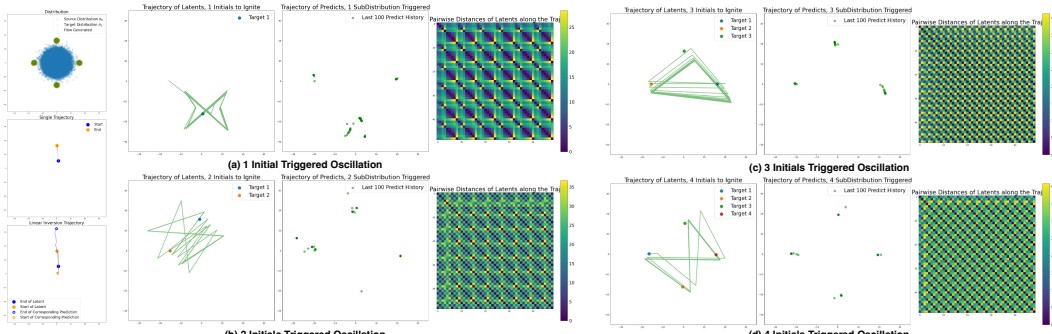

Figure 6: The oscillation phenomenon observed in the flow model trained on a toy distribution transitions from a large central Gaussian to a mixture of four smaller Gaussians. This behavior aligns closely with the results from our experiments on larger models.

In this section, we provide a theoretical framework to explain the oscillatory behavior observed in the fixed-point inversion of large-scale diffusion models, specifically within the rectified flow framework. We aim to show that, under certain assumptions and conditions, the inversion iteration will oscillate between several concentrated regions without diverging to unbounded, meaningless regions.

## 4 APPLICATIONS

In this section, we refer to our proposed method, Oscillation Inversion, as OInv. Our experiments consist of three parts: stroke-based human makeup synthesis to demonstrate the effectiveness of OInv-Finetune, human face post-enhancement to refine industry-standard enhancement results and verify OInv-Group, and finally, a quantitative evaluation of the reconstruction quality and guided diverse sampling compared to state-of-the-art inversion methods.

**Experiment Settings** All of our experiments are based on the 'black-forest-labs/FLUX.1-schnell' checkpoint. We run the experiments on a single A6000 GPU with 48GB of memory. All images are cropped and resized to $512 \times 512$ pixels. The oscillation inversion consistently runs for 30 iterations, taking 8.74 seconds per image. For OInv-Finetune, we fine-tune only the Attention modules of the model while freezing all other components, using a consistent 15-step process that takes 10.88 seconds in total. Once a suitable latent representation is found, the reverse steps take 3-5 iterations depending on the settings, and this step is optional.

### 4.1 IMAGE RESTORATION AND ENHANCEMENT

Image restoration and enhancement can be seen as a specialized editing task that aims to recover an underlying clean image with high fidelity and detail from degraded measurements. Existing inversion methods, such as BlindDPS Chihaoui et al. (2024), often achieve good perceptual quality but tend to compromise on fidelity to the original image. Meanwhile, current image restoration techniques and enhancers, like ILVR Choi et al. (2021), struggle with real-world blind scenarios where the type of degradation is unknown or undefined. Image enhancement is a challenging problem, as it involves transferring a degenerated distribution to a natural image distribution. To demonstrate the effectiveness of our method in discovering high-quality distributions, as discussed in Section **??**, we perform both qualitative and quantitative evaluations. We use real-world degraded (low-quality) images for the qualitative assessment and apply simulated noise, blur, and low-resolution degradation to the CelebA validation dataset ( Liu et al. (2015)) for the quantitative analysis. For the latter, we follow the protocols of previous studies, using metrics like PSNR and LPIPS to measure performance. We compare our approach against existing image restoration and enhancement methods, including BlindDPS, DIP, GDP, and BIRD ( Chung et al. (2023); Ulyanov et al. (2018); Fei et al. (2023); Chihaoui et al. (2024)). Since our approach is positioned as a post-processing image enhancement technique, we also select an image enhancement baseline. We chose Piscart due to its

strong identity preservation and efficient batch processing capabilities. To further demonstrate the efficacy of our method, we report not only the original performance of Piscart but also results from several straightforward diffusion-based editing techniques, such as FluxODEInversion and Flux Linear, which are counterparts to DDIM Inversion and SDEdit. Table 1 presents the quantitative results on the CelebA dataset, highlighting the superior fidelity of our method while maintaining excellent perceptual quality. Figure 7 provides visual examples of our method's ability to enhance and restore images in real-world blind scenarios, including cases of noise, blur, and other types of degradation. Our approach consistently restores richer details compared to existing open-source and commercial methods. During our experiments, we observed a recurring pattern of distribution oscillation. While it is challenging to determine which oscillation center provides the highest-quality results across different degradation types, the oscillation behavior remained consistent within each degradation recovery task. This observation suggests that our inversion method successfully introduces a distribution transfer mechanism. In large-scale experiments, we visually inspected a small subset of recovered samples from the three distribution centers and manually selected the best-performing center. This center index was then applied to the entire set of images within the same degradation recovery task, ensuring consistency across the dataset.

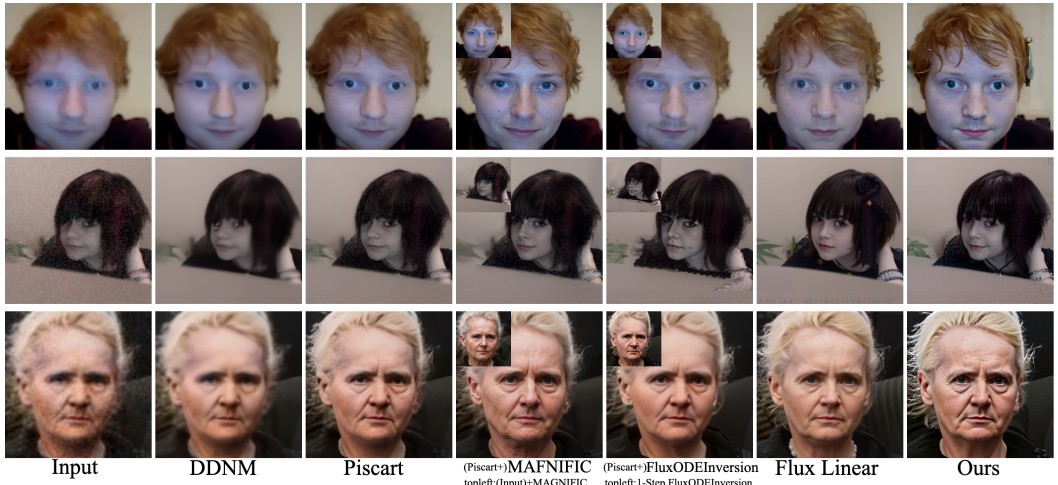

Input    DDNM    Piscart    (Piscart+)MAFNIFIC topleft:(Input)+MAGNIFIC    (Piscart+)FluxODEInversion topleft:1-Step FluxODEInversion    Flux Linear    Ours

Figure 7: Enhancer as Example of Triggered New Distribution from two Lower-quality Distribution

| Method | Denoise | | Deblur | | 4× SR | | Time (s) |
|--------|---------|-------|--------|-------|-------|-------|----------|
| | PSNR ↑ | LPIPS ↓ | PSNR ↑ | LPIPS ↓ | PSNR ↑ | LPIPS ↓ | |
| BlindDPS | - | - | 23.56 | 0.257 | 21.82 | 0.345 | 270 |
| DIP | - | - | - | - | 18.64 | 0.415 | - |
| GDP | - | - | 22.53 | 0.304 | 20.78 | 0.357 | 118 |
| BIRD | - | - | 24.67 | 0.225 | 22.75 | 0.306 | 234 |
| Piscart | **28.21** | 0.15 | **30.23** | 0.15 | **29.68** | **0.12** | 7.8 |
| (P)+ODEinv | 20.12 | 0.22 | 19.34 | 0.38 | 27.56 | 0.37 | +6.5 |
| (P)+Linear | 23.01 | 0.40 | 25.12 | 0.44 | 24.76 | 0.39 | **+4.5** |
| (P)+OInvOurs | 25.50 | **0.13** | 26.90 | **0.12** | 25.44 | 0.17 | +9.5 |

Table 1: Comparison of BlindDPS Chung et al. (2023), DIP Ulyanov et al. (2018) and GDP Fei et al. (2023) BIRD Chihaoui et al. (2024) on Denoise, Deblur, 4× Super Resolution (SR), and Time (in seconds). The best results are indicated in bold. + indicates the additional time required by the postprocessing method compared to its baseline time.

## 4.2 Low Level Editing

Our proposed method, oscillation group inversion with fine-tuning, is designed to support a variety of general low-level editing tasks. As illustrated in Figure 8, rough strokes from doodle drawings serve as guides for fine-tuning, while text prompts direct semantic details. Although our method can achieve effects like relighting and recolorization, these tasks are more challenging to evaluate due to the intuitive and non-precise nature of the prompts. Therefore, we primarily focus on makeup synthesis and transfer for validation. In the first part of our experiments, we use stroke prompts to create new makeup styles by altering facial and hair features. We compare our results with two recent makeup transfer methods, CSDMT (Sun et al. (2024)) and Stable-Makeup (Zhang et al. (2024)). In the second part, we take the 'before' and 'after' makeup images generated from the style transfer results as low-level prompts, using them to produce higher-quality makeup images. This demonstrates our method's ability to capture complex color distributions. Visual results are shown in Figure 8, and quantitative results are presented in Table 2. These results follow the metrics used by CSDMT for evaluation. The stroke editing experiments were conducted on a manually labeled dataset, which includes 10 high-quality face images and 77 different stroke variations, encompassing both makeup and hair color changes. The enhancement experiments were performed on the LADN dataset.

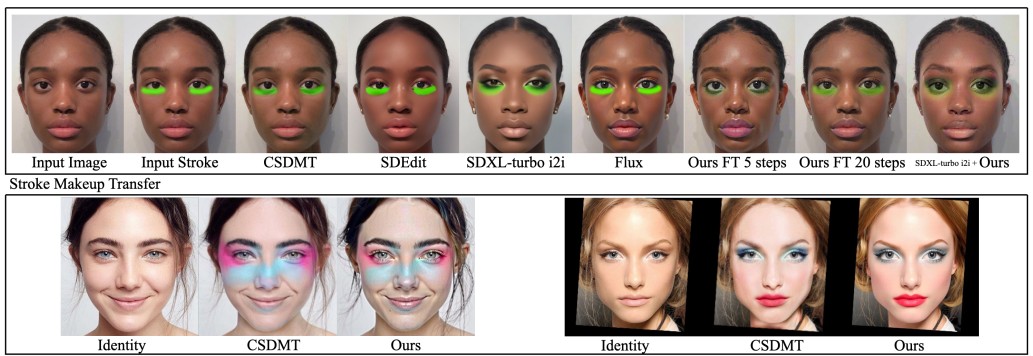

Figure 8: stroke makeup example and our enhanced result based on SOTA stylish make up transfer method CSDMT( Sun et al. (2024))

| Methods | PSNR Stroke | SSIM Stroke | PSNR Enhance | SSIM Enhance |
|---|---|---|---|---|
| SDEdit( Meng et al. (2021)) | 19.32 | 0.719 | 23.45 | 0.842 |
| SDXL-turbo i2i | 20.05 | 0.733 | 24.10 | 0.851 |
| Flux | 18.89 | 0.710 | 22.90 | 0.830 |
| Oinv Ours | **20.45** | **0.785** | 25.10 | **0.89** |
| CSDMT( Sun et al. (2024)) | 19.90 | 0.715 | **25.32** | 0.86 |
| Stable-Makeup( Zhang et al. (2024)) | 18.60 | 0.705 | 24.30 | 0.884 |

Table 2: Comparison of different methods on the task of stroke make-up transfer and enhancement. The metrics demonstrate that our method is robust in stroke makeup transfer and significantly improves the quality of the baseline method compared to others.

## 4.3 Reconstruction and Diverse Sample

We performed an evaluation on image reconstruction task on the COCO Validation set, utilizing the default captions as text prompts. The quantitative results are presented in Table 4.3. We followed the same settings in Pan et al. (2023b). We selected the existing inversion methods, including DDIM, NULL Text , AIDI, ReNoise ( Pan et al. (2023b); Mokady et al. (2023); Pan et al. (2023a); Garibi et al. (2024); Meiri et al. (2023b)) as competing methods. Our method achieves near-exact inversion, comparable to them.

Since our proposed inversion method is intended for low-level editing, we do not claim that it provides semantic editing capabilities. However, as discussed in Section 3.2.4, we explore optimization

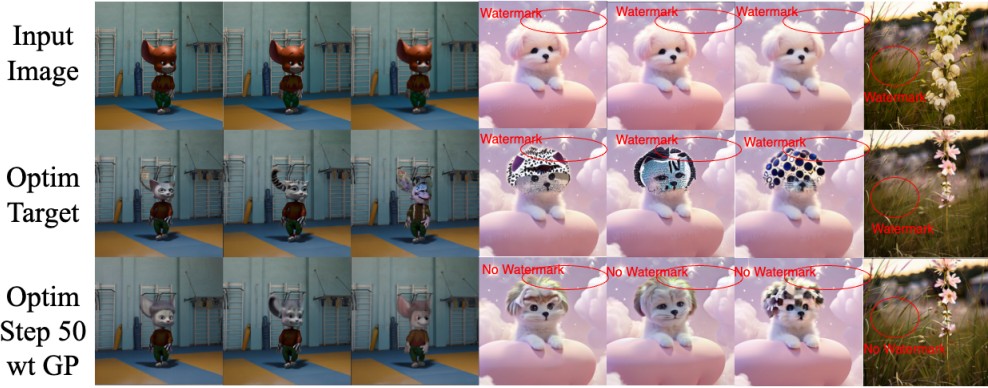

Figure 9: We use Image Space L2 Loss to align the latent with the visual prompt while preserving the image's style. Oscillation separates clean and watermarked domains, and optimization yields a watermark-free result, demonstrating the effectiveness of Gaussian Processing Regularization.

| Method | PSNR (dB) | LPIPS |
|--------|-----------|-------|
| DDIM | 12.20 | 0.409 |
| NULL Text | 25.47 | 0.208 |
| AIDI | **25.42** | 0.249 |
| EDICT | 25.51 | 0.204 |
| ReNoise | 17.95 | 0.291 |
| RNRI | 22.01 | 0.179 |
| Oinv (Ours) | 20.87 | **0.154** |

Table 3: Reconstruction Quality Comparison based on PSNR and LPIPS

| Method | CLIP | SSIM |
|--------|------|------|
| NULL Text | 0.68 | 0.80 |
| EDICT | 0.70 | 0.78 |
| IP-Adapter | 0.82 | 0.72 |
| MimicBrush | 0.60 | 0.84 |
| Oinv (Ours) | **0.85** | **0.87** |

Table 4: Editing Quality Comparison based on CLIP and SSIM

strategies around the inverted latents to achieve customized functions, as illustrated in Figure 9. Quantitative results are provided in Table 4.3

## 5 ABLATIONS

We also applied the same method to short sampling methods, including SD3 and the Latent Consistency Model; however, the same phenomena were not observed. Interestingly, we found this behavior to be quite prominent in our flow model trained from scratch on toy distributions.

## 6 CONCLUSIONS

In this work, we introduced Oscillation Inversion, a novel method for image manipulation within rectified flow-based diffusion models, addressing the challenges posed by the unique latent structures of these models. Our approach leverages oscillatory behavior in fixed-point iteration to enable semantic-based image optimization, while extensions like Group Inversion, Finetuned Inversion, and Post-Inversion Optimization provide flexibility for diverse image editing tasks. Theoretical analysis and extensive experiments across various applications, including image restoration, relighting, and watermark removal, validate the effectiveness of our method, showcasing improvements in both perceptual quality and fidelity.

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
