# OpenReview forum: "OscillationInversion: Understand the structure of Large Flow Model through the Lens of Inversion Method"
_ICLR.cc/2025/Conference — ICLR 2025 Conference Withdrawn Submission_

### Official Review · Reviewer_bPut · 2024-11-01

**Soundness:** 2
**Presentation:** 1
**Contribution:** 2
**Rating:** 3
**Confidence:** 4

**Summary:**

The paper describes a method a method called 'Oscillations Inversion' that allows one to recover the latent noise corresponding to an image in a rectified flow model. Using this method, the paper discovers the phenomena that repeated repeated mapping and inversion between the latent representation and the pixel space images does not converge to a fixed point, rather it oscillates between distinct clusters.

Then, the paper proposes a method for guiding this oscillation using finetuned inversion. This refers to finetuning the flow model to align the velocity field of an inpainted image with the original image. This finetuning procedure is used to perform high-quality inpaining.

Finally, the paper proposes a method for further quality enhancement: post-inversion optimization. This enables the optimization of the latent representation to improve image fidelity metrics.

The experimental results show good performance on image inpainting and image enhancement tasks.

**Strengths:**

The paper showcases a very interesting finding in that if one uses the proposed inversion method, the image and their latent representation oscillates between different clusters.

The paper proposes two further methods, finetuned inversion and post-inversion optimization that are applied to tackle denoising and image inpainting tasks. The paper showcases impressive results on these tasks.

**Weaknesses:**

The oscillatory behavior in the fixed point iteration is induced by the error in the approximate inversion. A flow model defines a 1-to-1 mapping between the noise representations and latent representations. If the inversion process were exact, the process would not oscillate. It would map back and forth between the noise representation and the latent representation. Therefore the finding of the oscillatory behavior is not a general finding about flow models, rather its something specific to this inversion process.

The paper does not provide sufficient evidence for its central claim: The images and their latent representation oscillate between local clusters. The paper claims this is the case empirically, but provides no experimental data. The claim also isn't justified theoretically or an intuitive reasoning given why that would be the case.

Lack of justification for methods. It is not clear where the training objectives come from and why the methods achieve the desired results.
* Finetuned inversion: It is not clear why this training objective would induce more diverse oscillatory behavior in latent space. The paper lacks justification or experimental data that supports this claim.
* Post-inversion optimization: It is unclear why this complex training objective is chosen over a simple fidelity loss with a MSE regularization term.

Paper formatting:
* Figure 6 is not legible.
* Page 5 margin violation
* Top of Page 6 odd figure spacing?
* Table formatting does not follow ICLR convention.

**Questions:**

* Is the GP loss in Eq 9 equivalent to MSE? if z_i is fixed, the first and last terms are constant. The middle terms for an RBF kernel is simply the squared distance so this loss should be equivalent the MSE.

* In post-inversion optimization: How are the points assigned to clusters? How are the clusters identified?

---

### Official Review · Reviewer_Bmmb · 2024-11-03

**Soundness:** 2
**Presentation:** 1
**Contribution:** 1
**Rating:** 3
**Confidence:** 4

**Summary:**

The paper studied the oscillation problem in the inversion problem of rectified flows. The authors use a numerical method to justify the claim that oscillations exist as groups. Then they introduced finetuned inversio to make the separated clusters to align with customized semantics. The observation in the paper is largely expereiment based and the method introduced is strainghtforward.

**Strengths:**

The oscillation phenomen in the inversion problem of rectified flow is identified numerical. A solution to uitlize the oscillation is proposed to improve the quality of  image generations.

**Weaknesses:**

The main weakness is that the observation is identified through experiments not analysis. It is hard to fullly make the readers convinced of their contribution. The optimization method to deal with oscillation  is strainghtforwad without further explanation.

**Questions:**

1.	The reference format in section 2 looks very strange. Should be updated.
2.	As the paper is a direct follow up of paper Liu et al 2022, it is much better to elaborate the formulation in section 3.1.1.
3.	In (2), the integral on t is missing compared to Liu et al 2022.
4.     The lines 270-272 are not correctly displayed
5.     In line 370, "Section ??"

---

### Official Review · Reviewer_5nuJ · 2024-11-03

**Soundness:** 3
**Presentation:** 2
**Contribution:** 3
**Rating:** 5
**Confidence:** 2

**Summary:**

The paper propose a method for solving the image inversion problem. The authors observe that when using a fixed point iterative method to solve the inversion problem, the solutions oscillates number of points. Based on this observation a method for image edit, image reconstruction, and super resolution is derived.

**Strengths:**

1. The suggest method for solving the inversion problem which is of interest for a large audience.
2. The observation of oscillatory behavior for fixed point method on flow models and it relation to semantic meaning is novel. And the author are able to reproduce this behavior on smaller model as well.
3. The method is able to produce SOTA results.

**Weaknesses:**

1. No clear algorithm for applying the method is provided. This makes it hard to understand the order of which the 3 stages of the method, Group Inversion, Finetuned inversion, and Post-inversion Optimization is applied, or whether all of the three stages are always applied.

2. It is not clear how the use of multiple latent encoding in equation 6. is used for image reconstruction and image super resolution.

3. In equation 8 appears $\mathcal{L}_{\text{rgb}}$ and its said to be "a customized loss function" but no example of such loss function is given.



4. images in figure 6 are too small

**Questions:**

Can the authors please expand on weaknesses 1 and 2.

---

### Official Review · Reviewer_e8X8 · 2024-11-04

**Soundness:** 3
**Presentation:** 3
**Contribution:** 3
**Rating:** 6
**Confidence:** 3

**Summary:**

The paper presents a novel method called Oscillation Inversion for enhancing image manipulation capabilities in rectified flow-based large text-to-image diffusion models. The authors observe that, unlike conventional inversion methods that converge smoothly, their proposed approach oscillates between distinct semantic clusters.

**Strengths:**

- The method's applicability to diverse tasks (e.g., image restoration, enhancement, and makeup transfer) demonstrates its flexibility and potential for real-world uses.

**Weaknesses:**

- Are there other models to perform the experiments with? The experiments are just on one checkpoint. (FLUX)

**Questions:**

See the weaknesses.

---

### Note · Authors · 2024-11-15

I have read and agree with the venue's withdrawal policy on behalf of myself and my co-authors.